

# Opinion: A Critical Evaluation of the Evidence for Aerosol Invigoration of Deep Convection

Adam C. Varble[1], Adele L. Igel[2], Hugh Morrison[3], Wojciech W. Grabowski[3], and Zachary J. Lebo[4]

[1]Atmospheric Sciences and Global Change Division, Pacific Northwest National Laboratory, Richland, WA, USA
[2]Department of Land, Air, and Water Resources, University of California, Davis, Davis, CA, USA
[3]National Center for Atmospheric Research, Boulder, CO, USA
[4]School of Meteorology, University of Oklahoma, Norman, OK, USA

*Correspondence to*: Adam C. Varble (adam.varble@pnnl.gov)

**Abstract.** Deep convective updraft invigoration via indirect effects of increased aerosol number concentration on
cloud microphysics is frequently cited as a driver of correlations between aerosol and deep convection properties.
Here, we critically evaluate the theoretical, modeling, and observational evidence for warm- and cold-phase
invigoration pathways. Though warm-phase invigoration is plausible and theoretically supported via lowering of the
supersaturation with increased cloud droplet concentration in polluted conditions, the significance of this effect
depends on substantial supersaturation changes in real-world convective clouds that have not been observed. Much of
the theoretical support for cold-phase invigoration depends on unrealistic assumptions of instantaneous freezing and
unloading of condensate in growing, isolated updrafts. When applying more realistic assumptions, impacts on
buoyancy from enhanced latent heating via fusion in polluted conditions are largely canceled by greater condensate
loading. Foundational observational studies supporting invigoration have several fundamental methodological flaws
that render their findings incorrect or highly questionable. Thus, much of the evidence for invigoration has come from
numerical modeling, but different models and setups have produced a vast range of results. Furthermore, modeled
aerosol impacts on deep convection are rarely tested for robustness, and microphysical biases relative to observations
persist, rendering many results unreliable for application to the real world. Without clear theoretical, modeling, or
observational support, and given that enervation rather than invigoration may occur for some deep convective regimes
and environments, it is entirely possible that the overall impact of cold-phase invigoration is negligible. Substantial
mesoscale variability of dominant thermodynamic controls on convective updraft strength coupled with substantial
updraft and aerosol variability in any given event are poorly quantified by observations and present further challenges
to isolating aerosol effects. Observational isolation and quantification of convective invigoration by aerosols is also
complicated by limitations of available cloud condensation nuclei and updraft speed proxies, aerosol correlations with
meteorological conditions, and cloud impacts on aerosols. Furthermore, many cloud processes such as entrainment
and condensate fallout modulate updraft strength and aerosol-cloud interactions, varying with cloud life cycle and
organization, but these processes remain poorly characterized. Considering these challenges, recommendations for
future observational and modeling research related to aerosol invigoration of deep convection are provided.



## 1 Introduction

There are many proposed effects of aerosols on deep convection. Cloud microphysical effects include the dependence
of cloud droplet number concentration ($N_d$) and size on the number of cloud condensation nucleating (CCN) aerosols
at a given water vapor supersaturation (Squires 1957, Twomey and Squires 1959, Squires and Twomey 1960). Such
an effect modulates the efficiency of rain formation via the collision-coalescence process, which is suppressed as CCN
number concentration increases with all else being constant including warm cloud depth, cloud base temperature,
updraft speed, and updraft dilution (Gunn and Phillips 1957, Rosenfeld 1999). In deep convective clouds, droplets
grow as they ascend in updrafts where they can be vertically transported to subfreezing temperatures. At temperatures
below 0°C, several factors influence whether enough ice initiates and grows to glaciate the cloud, including the
temperature, CCN-modulated droplet size distribution, concentration of ice nucleating particles, and secondary ice
production (Cantrell and Heymsfield 2005). Glaciation further affects the cloud's precipitation efficiency and the
micro- and macro-physical properties of stratiform anvil clouds that exert a dominant control on the net radiative
effects of deep convective clouds (Feng et al. 2011, Gasparini et al. 2019). The CCN-modulated supercooled liquid
drop size distribution can also influence the riming and secondary ice production rates in the mixed phase portion of
convective updrafts (Korolev et al. 2017), which exerts a strong control on the non-inductive charging of ice particles
that supports much of the lightning in deep convection (Takahashi 1978).

Aerosol effects have been proposed that influence the cloud dynamics, convective updrafts in particular, extending
beyond direct changes to deep convective cloud microphysical processes and properties. One such mechanism
operates via changes in atmospheric thermodynamic stability induced by aerosol enhancement of the scattering or
absorption of shortwave radiation (Ackerman et al. 2000, Koren et al. 2004). Additional proposed mechanisms are the
invigoration of deep convection due to increased latent heating by condensation in warm clouds and fusion in cold
clouds with increased CCN number concentrations (see many studies listed in Igel and van den Heever 2021).
"Invigoration" in this context refers to an increase in the vertical wind speed of convective updrafts. This follows from
the definition of convective intensity that is often based on updraft speed and is consistent with studies that laid the
foundation for the theory of deep convection invigoration by aerosol. A clear definition of invigoration is important
because many studies have conflated invigoration with changes to other deep convective cloud properties, such as
radar reflectivity, precipitation, and lightning flash rate, that may or may not be associated with a change in the updraft
speed. Aerosol-induced convection invigoration is viewed as potentially important for climate because CCN
concentration is impacted by anthropogenic emissions, and there is the potential for convective intensity changes to
alter convective vertical transport, precipitation, and radiative effects.

This paper specifically focuses on evaluating the evidence for aerosol invigoration of deep convection driven by
enhanced latent heating, and thus increased cloud buoyancy, since numerous studies have invoked it to explain
correlations between aerosol and convective cloud properties. We specifically assess two mechanisms theorized to
drive convective invigoration with increased aerosol loading: 1) mixed-/cold-phase invigoration whereby higher CCN
and thus droplet concentrations suppress warm-rain production, leading to greater lofting of cloud condensate mass
and increased fusion heating when the droplets freeze; 2) warm-phase invigoration whereby higher CCN and thus



cloud droplet number concentrations increase condensation heating. Several papers have also been published recently
that contradict the studies that laid the groundwork for these theories. We provide overviews and critical evaluations
of the theoretical, modeling, and observational foundations of deep convection invigoration by aerosols before ending
with some concluding thoughts on a path forward to improving research, understanding, and quantification of aerosol
impacts on deep convective clouds.

## 2 Theoretical Foundation

Aerosol-induced deep convection invigoration was most prominently discussed by Rosenfeld et al. (2008), hereafter
R08. According to this paper, invigoration is a multistep process that can be understood by viewing convection as a
rising parcel of air. First, enhanced CCN concentrations lead to an increase in droplet number concentration that
suppresses warm-rain production, thereby increasing the condensed water mass. When this more heavily laden parcel
is subsequently lifted above the 0°C level, additional latent heat can be released via freezing of the greater condensed
water mass. The freezing compensates the loss of buoyancy associated with extra condensate carried across the
freezing level. Freezing simultaneously induces precipitation, reducing the condensate loading to yield a boost in
buoyancy. This extra buoyancy is manifested as an increase of up to 1000 J kg$^{-1}$ in available potential energy in the
R08 example if all condensate is immediately frozen and removed from the parcel, which is available to increase
vertical velocity. The R08 invigoration pathway is very often explained simply as the result of suppressed precipitation
leading to more freezing, but we stress here that this invigoration theory critically hinges on the assumption that all
liquid freezes quickly at a relatively warm temperature and that condensate is unloaded upon freezing (Grabowski and
Morrison 2020; Igel and van den Heever 2021).

The R08 theory, referred to as "mixed-" or "cold-phase invigoration", is well known (at least in the simplified form)
and often used as an explanation for correlations between aerosol metrics and convective cloud properties, as
evidenced by its 1340+ and growing citations (Web of Science, April 2023). However, this theory has been critically
examined by several studies in the 15 years since its publication, and multiple lines of evidence from theory, modeling,
and observations suggest that it is not a major contributing factor to aerosol-induced invigoration. Igel and van den
Heever (2021) directly evaluated the original cold-phase invigoration theory presented in R08. They point out that the
original calculations to support the theory make several major and unrealistic assumptions, in particular regarding
instantaneous and complete unloading and freezing of condensate. They ran a suite of new calculations with updated,
more realistic assumptions, though still in the context of parcel theory, to show that the cold-phase invigoration
mechanism proposed by R08 is at best weakly positive but also potentially weakly negative. Examples of the major
impacts that unloading and freezing assumptions have on updraft parcel density temperature are shown in Figure 1.
Weak positive or negative impacts on updrafts are supported by numerical simulations discussed in Grabowski and
Morrison (2016, 2020), Lebo (2018), and Heikenfeld et al. (2019). Parcel calculations show that the inclusion of
additional processes such as entrainment further weaken cold-phase invigoration, potentially making updraft
weakening more probable than strengthening in response to precipitation suppression (Peters et al. 2023, in review).



A second major theory has emerged since R08. This theory, referred to as "warm-phase invigoration", postulates that a polluted rising parcel with a higher cloud droplet number concentration caused by higher CCN number concentration will condense water faster and lower the supersaturation within the parcel (e.g., Lebo et al. 2012, Koren et al. 2014, Fan et al. 2018, Cotton and Walko 2021). This additional latent heating increases the buoyancy of the parcel and gives rise to higher vertical velocities. The theory can be traced to Kogan and Martin (1994) and many others since then, including Fan et al. (2007), Grabowski and Jarecka (2015), and Igel and van den Heever (2021). An important caveat is that the condensation rate (and thus the rising adiabatic parcel latent heating) only depends on the parcel ascent rate when the local supersaturation can be assumed equal to its quasi-equilibrium value (see section 2b in Grabowski and Morrison 2020). The quasi-equilibrium supersaturation represents an exact balance between the supersaturation source due to parcel ascent and the supersaturation sink due to droplet population growth (Politovich and Cooper (1988) and references therein; see also the appendix in Grabowski and Morrison (2021)). In the case of quasi-equilibrium supersaturation, the only possible invigoration of the cloud updraft by CCN concentration comes from the quasi-equilibrium supersaturation being smaller when the droplet concentration increases, in effect increasing the adiabatic cloud updraft buoyancy.

Whether this theory is of practical importance hinges on the magnitude of supersaturation that can be achieved in convective storms and whether the quasi-equilibrium supersaturation provides an accurate approximation to the in-cloud supersaturation. Politovich and Cooper (1988) argued for the validity of the quasi-equilibrium supersaturation approximation, except near cloud base. One should also expect that the quasi-equilibrium approximation should not apply to volumes strongly affected by rain washing out a large fraction of the cloud droplet population. Based on updraft parcel calculations, Figure 1 shows that supersaturation differences between clean and polluted conditions of 5% or more would be required to produce notable impacts on updraft buoyancy. In idealized bin microphysics simulations, Hall (1980) and Lebo et al. (2012) showed supersaturation values exceeding 5% and argued that they originated from removal of cloud water by precipitation processes and an inability of the remaining cloud droplets to take up the available water vapor by diffusional growth in the strong cloud updrafts. In numerical simulations discussed in Grabowski and Morrison (2016, 2020), supersaturation values close to 10% occurred, especially in very low CCN conditions, with noticeably stronger updrafts below the freezing level produced in high CCN simulations. While supersaturation in convection cannot currently be directly measured, some observational inferences suggest that the supersaturation is rarely sufficiently high for notable warm-phase invigoration (e.g., Politovich and Cooper 1988, Prabha et al. 2011, Romps et al. 2023). On the other hand, some modeling studies suggest that the supersaturation can easily be high enough (Khain et al. 2011, Grabowski and Morrison 2021), but their location in the vertical at high levels may limit the magnitude of invigoration (Lebo et al. 2012). Unlike cold-phase invigoration, warm-phase invigoration is always positive and does not rely on unfounded assumptions. Thus, it more plausibly explains aerosol-induced convective updraft invigoration. However, whether such invigoration is sufficiently great to be observationally detectable and consequential to weather and climate remains unknown.

The above discussion is simplified over real-world deep convection in that it neglects variability in critical cloud dynamical and microphysical processes that control condensation, freezing, and condensate loading, as well as their



dependence on meteorological conditions and updraft properties. It also assumes that convective updrafts are separated

from other updrafts and clouds. However, deep convection organized into multi-cell mesoscale systems contributes most of the convective precipitation globally (Nesbitt et al. 2006, Roca et al. 2014, Feng et al. 2021), and updrafts in such systems are affected by additional processes such as interactions of updrafts with pre-existing clouds, gravity waves, and cold pool outflows that are not considered in warm- and cold-phase invigoration theories. They also do not account for potentially longer- and larger-scale adjustments of environmental conditions to changes in convective

heating that can buffer any potential invigoration effects. Such processes are very difficult to treat theoretically, and thus for complex, real-world convective cloud situations, model simulations with observational validation must be relied upon to advance understanding.

**3 Modeling Foundation**

Atmospheric models have been a key tool for studying aerosol impacts on deep convection, including invigoration.

This has typically been done using nonhydrostatic models that can explicitly represent deep convection with a grid length between 1 and 5 km, though they poorly resolve individual convective updrafts and entrainment. Such models were used to study deep convection in the 1980's and 1990's, often referred to as cloud-resolving models, and more recently as convection-permitting or convection-allowing models. In the 21st century, they have been widely used for operational numerical weather prediction. Advances in computing power have also made it possible to simulate deep

convection with higher-resolution large-eddy simulation (LES) models that have grid lengths of a few hundred meters or less (e.g., Bryan et al. 2003; Khairoutdinov et al. 2009). These models have a significant advantage over lower-resolution cloud models because they can better resolve individual deep convective updraft properties (including width and strength) and large turbulent eddies (Bryan et al. 2003; Lebo and Morrison 2015). LES has become widely adopted in the past 10 years for simulating deep convection, particularly for idealized studies.

For quantifying aerosol impacts on clouds, models have a major advantage compared to observations because sensitivity experiments can be performed with altered aerosol characteristics (e.g., increased aerosol loading) while keeping all other aspects of the model the same. Following this approach, many cloud modeling studies have shown invigoration of deep convective updrafts with increased aerosol number concentrations (e.g., Khain et al. 2004, 2005, 2012; Wang 2005, van den Heever et al. 2006, Seifert and Beheng 2006, Fan et al. 2007, 2013, 2018; Storer et al.

2010, Storer and van den Heever 2013, Chen et al. 2017, 2020). These papers have often been cited as supporting cold-phase invigoration as described by R08 or have themselves made this claim (e.g., Fan et al. 2009, 2012). A similar situation pertains to studies on warm-phase invigoration (e.g., Sheffield et al. 2015, Fan et al. 2007, 2018; Cotton and Walko 2021). Although the end impacts on simulated convective updrafts are attributable to aerosols using the approach of model sensitivity studies with perturbed aerosol loading, the process-level interpretation is often

muddied. In particular, specific mechanisms driving changes in convective intensity with aerosol changes can be difficult to isolate with even detailed analysis because of the complex interactions and feedbacks between various microphysical and dynamical processes across scales. For example, changes in the environment driven by aerosol loading may intensify updrafts (for example, through cold pool-convection interactions, cf. Lebo and Morrison 2014).





In turn this will increase the latent heating rate owing to the close connection between updraft vertical velocity and
condensation rate. Thus, in this situation, increased latent heating occurs with stronger updrafts, but it is not the
primary explanation for why the updrafts are stronger. Because virtually any change in updraft strength is accompanied
by changes in latent heating, attribution of aerosol impacts to specific microphysical process pathways can be very
challenging. This is akin to a "chicken-and-egg" problem – what comes first, changes in updraft intensity or changes
in latent heating? A few studies have attempted to directly test invigoration mechanisms by modulating process rates
explicitly (Seiki and Nakajima 2014, Nishant et al 2019, Abbott and Cronin 2021), although these results can still be
inconclusive for reasons discussed below.

Besides the difficulty with process attribution, there are several additional factors that contribute to a lack of clarity
regarding modeling aerosol impacts on deep convection. First, models are imperfect and suffer from numerous
uncertainties and biases. These include uncertainties in physical parameterizations and their coupling with the model
dynamics, the inability to resolve the full range of turbulent and cloud-scale motions (particularly in lower-resolution
cloud models), and uncertainties in initial and lateral boundary conditions. Since the mechanism of aerosol
invigoration involves CCN effects on cloud and precipitation particles that in turn impact the cloud dynamics, the
parameterization of cloud microphysics in models is an especially critical link.

Unfortunately, many aspects of parameterizing microphysics remain highly uncertain (e.g., Morrison et al. 2020).
There are two main drivers of this uncertainty. The first is uncertainty in how the multitude of cloud and precipitation
particles are represented, as it is impossible computationally to explicitly simulate every particle in a cloud. Various
approaches have been taken to address this problem (Khain et al. 2015; Grabowski et al. 2019), including 1)
computationally efficient bulk schemes that predict only one or a few bulk quantities of the particle population such
as water content and number concentration; 2) bin schemes that explicitly evolve particle populations by dividing
them into size or mass bins; and 3) Lagrangian particle-based schemes that represent the particle population with a
limited number (typically ~100 per grid volume) of computational particles that are tracked in the modeled flow (called
"super-droplets" or "super-particles"), each representing some multitude of real particles. Note that Lagrangian
particle-based schemes with both liquid and ice super-particles are in their infancy; hence, the use of these schemes
so far has been very limited for modeling deep convection, but this is anticipated to change within the next 5-10 years.

The second major source of uncertainty is limited fundamental knowledge of cloud physics, especially for ice-phase
processes. Particularly relevant to "cold-phase invigoration" of deep convection, there is large uncertainty in how ice
particles are produced, grow through various processes, and fall out (Morrison et al. 2020). This includes secondary
ice production, which is the generation of new ice particles through mechanisms other than primary ice nucleation
(heterogeneous nucleation on ice-nucleating particles or homogeneous nucleation within drops). Secondary ice
processes are currently highly simplified in models but may be crucial for some types of convective clouds (e.g., Field
et al. 2017, Korolev and Leisner 2020). While ice microphysics is particularly uncertain, aspects of warm (liquid)
microphysics remain uncertain as well, especially the problem of drop collision-coalescence and breakup.
Consequently, there is considerable uncertainty in how microphysics is represented in all cloud models, even those
running the most sophisticated bin and Lagrangian microphysics schemes. For instance, a recent intercomparison of



bulk, bin, and Lagrangian schemes in simple box and 1D models showed convergence for the Lagrangian schemes
considering only the problem of droplet activation and condensation, but these schemes diverged when collision-
coalescence was included (Hill et al. 2023). Because there is limited knowledge of the underlying microphysical
processes, a wide variety of process formulations are employed in different models. This has contributed to a wide
spread of model results for the same deep convection cases (e.g., Varble et al. 2011, Fridlind et al. 2012, Zhu et al.

2012, Varble et al. 2014a-b), even using the same model with the only change being the microphysics scheme (e.g.,
van Weverberg et al. 2012, Stanford et al. 2017, Han et al. 2017, Fan et al. 2017, Xue et al. 2017). In this context, it
is not surprising that the spread of simulated aerosol impacts on deep convective clouds is large among different
models and parameterizations (White et al. 2017, Marinescu et al. 2021).

Another issue is the representativeness and robustness of model simulations of deep convection invigoration. A key
consideration is the model configuration. Many studies of aerosol impacts on deep convection, including most early
studies using bin microphysics, modeled isolated single storms using small domains (order few tens of km) over time
periods up to several hours. These studies also typically used idealized boundary conditions, which can strongly
modulate aerosol effects on convective clouds (Dagan et al. 2022). With *open* lateral boundary conditions, the flux of
water vapor into the domain is not constrained, and thus large changes in cloud dynamics and precipitation can occur

with aerosol modification. On the other hand, models with *periodic* lateral boundary conditions and small domains
cannot capture interactions between convection and the larger-scale dynamics, including impacts on cold pools given
that the cold pool is confined within the model domain in this type of setup. It is likely that applying different boundary
conditions and microphysics schemes, as well as simulating different cases, have contributed to the large spread of
aerosol impacts on deep convection reported in the literature (e.g., from -93% to +700% change in surface precipitation

in the review paper of Tao et al. 2012). Some studies have also highlighted a dependence of simulated convective
invigoration on environmental conditions, namely vertical wind shear and free tropospheric relative humidity (Khain
2009, Fan et al. 2009, Lebo and Morrison 2014, Grant and van den Heever 2015). Additional spread in simulated
aerosol impacts may simply be due to different flow realizations, as discussed below.

For models with larger domains (100 or more km wide) containing numerous clouds interacting over longer periods
(~12-24 hours or more), convective invigoration is constrained by feedbacks between convection and its larger-scale
environment. For example, under steady, horizontally uniform forcing, the environmental thermodynamic stability
and moisture adjusts to aerosol-induced convective invigoration; subsequently, convection returns to its original
intensity, which is determined by the forcing (Morrison and Grabowski 2013). The timescale for this adjustment is
controlled approximately by the mean cloud spacing and gravity wave speed. A similar idea holds under less idealized

conditions, where the invigoration of updrafts and increased precipitation from one cloud or region of clouds will lead
to greater stability and less water available for other clouds and cloud regions, all else being equal. Thus, invigoration
could be short-lived and/or localized for individual convective events, but caution should be exercised in interpreting
any such impacts as a sustained phenomenon. Consistent with this idea, Siefert et al. (2012) showed little net change
in precipitation by uniformly increasing droplet concentration in the domain (as a proxy for aerosol loading) in a series

of 48-hour simulations using a weather forecasting model. Although there is limited evidence for convective



invigoration when aerosol properties are modified throughout the model domain, horizontal gradients of aerosol properties could drive invigoration over a limited region (i.e., smaller than the Rossby radius), sustained by circulations that develop between polluted and pristine regions (Morrison and Grabowski 2013, Leung and van den Heever 2023). In idealized simulations applying the weak temperature gradient (WTG) approximation to parameterize

large-scale ascent, Abbott and Cronin (2020) showed an increase in free tropospheric relative humidity with increased cloud droplet concentration due to greater detrainment and mixing of cloud condensate. This led to less dilution of subsequent clouds, greater large-scale ascent, and stronger convection in a positive feedback, without involving the warm-phase or cold-phase invigoration mechanisms.

Even for situations in which convective invigoration may be expected, limited predictability and the impact of different

flow realizations are critical to consider in analyzing model simulations. A fundamental behavior of atmospheric flow models is the rapid growth of initially small perturbations, both in amplitude and scale. Tiny initial differences between two simulations often lead to substantial divergence between the simulations at convective scales within a few hours. This divergence can make it difficult to know if differences between two simulations run with different aerosol conditions are robust. This is a particular concern for "real case" model setups with realistic forcing and initial and

lateral boundary conditions. It is also relevant for idealized models given the sensitivity of aerosol impacts on deep convection to small changes in initial conditions and forcing (e.g., Morrison 2012, Grabowski 2018). Averaging (in space and/or time) can help to address this issue, but effects are expected to be "washed out" as the spatial or temporal averaging scale is increased for the reasons explained above.

One approach to improving robustness is to run ensembles and compare two or more sets of simulations having

different aerosol conditions (Miltenberger et al. 2018). By calculating ensemble spread within each set, statistical significance of aerosol impacts – the difference between sets – may be determined. Another approach is to employ microphysical "piggybacking", which has been combined with small (3-5 member) ensembles in past studies (e.g., Grabowski 2014, Grabowski and Morrison 2017, Grabowski 2019, Sarkadi et al. 2022). In this approach, the model dynamics are coupled to one set of thermodynamic and cloud variables in a two-way feedback, while a second set

(e.g., with modified aerosol) is driven by the flow but does not feed back to it. This allows for point-by-point assessment of aerosol impacts on microphysics and cloud buoyancy for the same flow field. Moreover, in modeling studies, it is often difficult to assess the mechanisms that drive invigoration because of complicated interactions and feedbacks between the microphysics and dynamics noted above, and piggybacking can help to address this problem. By separating thermodynamic and dynamic drivers, piggybacking can help to isolate these mechanisms. That said,

the piggybacking methodology has drawn criticism from invigoration proponents (see comments in Fan and Khain (2021) and responses in Grabowski and Morrison (2021)).

To briefly summarize, models have substantial biases and uncertainties that impact their ability to simulate cloud-aerosol interactions and convective invigoration in particular. A focus on improving models, particularly how they treat cloud microphysical processes, is needed to reduce uncertainty and increase confidence in numerical studies of

convective invigoration. However, even in the hypothetical situation of having a perfect model there would still be challenges in interpreting results, and well-designed experiments are critical for a robust assessment of convective



invigoration. In particular, the rapid growth of small perturbations at convective scales implies a need for ensembles to quantify uncertainty rigorously, especially for "real case" simulations. Moreover, given that aerosol effects on convective clouds vary across different convection regimes, there is a challenge of generalizing over a range of conditions. Overall, this leads to the conclusion that a large amount of model data over many cases is needed to obtain robust results, while also considering that models are imperfect and often substantially biased.

## 4 Observational Foundation

Observational studies are critical for assessing the accuracy of modeling results and their applicability to reality. One of the first prominent observational studies to hypothesize the potential for increased aerosol loading to invigorate updrafts via suppressed coalescence-driven precipitation was Williams et al. (2002) who analyzed relationships between lightning, convective available potential energy, and aerosols in the Amazon. This was followed by Andreae et al. (2004) concluding that convective clouds were more dynamically vigorous in smoky regimes relative to clean regimes over the Amazon due to suppressed precipitation leading to enhanced latent heating from fusion. These studies inferred potential invigoration of updrafts from substantial cloud microphysical changes without direct evidence of updraft strength changes. Several prominent studies followed (e.g., Koren et al. 2005, Li et al. 2011) that claimed to show clearer observational evidence of deep convective updraft cold-phase invigoration by aerosols, but methodological and inferential limitations and flaws in those studies call such a conclusion into question, as discussed further below. This consideration is important because such studies laid the foundation for numerous observational studies since that repeated some methodological flaws of these early studies (e.g., Altaratz et al. 2010, Koren et al. 2010, 2012; Yuan et al. 2011, Niu and Li 2012, Storer et al. 2014, Yan et al. 2014, Guo et al. 2018; Liu et al. 2018, Hu et al. 2019, Pan et al. 2021) or relied on them to infer causal mechanisms for relationships between aerosol and deep convection properties (Lin et al. 2006, Guo et al. 2016, Thornton et al. 2017, Jiang et al. 2018, and many more).

The first major limitation of studying aerosol interactions with convective updrafts is a scarcity of routine CCN (even for a constant supersaturation) and updraft speed measurements over a range of conditions, and even fewer examples of co-located CCN and updraft measurements. Thus, proxies for CCN and updraft speed are often required to generate sampling that is sufficient for generating statistical relationships. Commonly used proxies for updraft base CCN are surface-based measurements of condensation nuclei (CN), CCN, particulate matter, aerosols within a particle size range, or aerosol optical depth (AOD), while satellites further provide AOD over much larger regions. A shortcoming of discrete surface sites is that they require convective clouds to form within sufficient range and direction for surface measurements to influence cloud inflow, which greatly limits sampling. The primary issues with AOD are that it does not always correlate with low-level CCN (Stier 2016, Veals et al. 2022) and can only be measured for clear skies. An example of this issue is highlighted in Figure 2 using simulation output where clouds block AOD retrievals in the inflow near the strongest updrafts, while surface aerosol concentration varies substantially by location and does not correlate with AOD near clouds. AOD has also been shown to increase with relative humidity (RH) due to aerosol water uptake (Quaas et al. 2010, Chand et al. 2012), and RH is commonly higher near clouds. Clouds can also alter AOD via cloud contamination of perceived clear skies, 3D cloud radiative effects, detrainment of cloud-processed



aerosols and moisture, and possible cloud-induced new particle formation (Marshak et al. 2021). Since deep convective clouds commonly form along or near boundaries separating distinctive air masses with updraft inflow coming from a specific direction, and because convective outflows at the surface and aloft alter aerosol properties, the location at which AOD is sampled is important, but this is rarely considered in studies. Sampling locations and times are similarly important when surface site measurements are used. Öktem et al. (2023) showed that the conclusion of warm-phase invigoration in Fan et al. (2018) was not robust if objective aerosol sampling was applied. Whether deep convection is surface-based or fueled by air elevated off the surface also impacts the relevance of surface aerosol measurements and needs to be considered in analyses. Rosenfeld et al. (2016) presented a technique for deriving cloud base CCN concentration from satellite-retrieved cloud droplet number and a simple cloud base updraft speed parameterization for non-raining, unobscured boundary layer convective clouds. This is a good step toward increasing the number of CCN retrievals, though care is still required in usage and interpretation of such retrievals given their limited validation and application to select situations that are not necessarily representative.

Vertical wind speed retrievals in deep convection are also rare. The most accurate retrieval of updraft speed is via aircraft (e.g., see Lucas et al. (1994) and references therein) but such penetrations are not common, are often not representative, and lack spatiotemporal context. Vertically profiling radars provide more context with slightly lesser accuracy (e.g., see Giangrande et al. (2016) and references therein) due to imperfect hydrometeor fall speed corrections, but similarly suffer from limited sampling and often missing the most intense portion of updrafts because updrafts often shear horizontally and need to pass over the profiler. Multi-Doppler scanning radar vertical velocity retrievals provide spatial structure and time evolution (e.g., see North et al. (2017) and references therein), but again suffer from deficient sampling. Such retrievals typically have limited spatial resolution (> 1 km) due to time integration, beam spacing, and smoothing with uncertainties of several m/s such that isolation of aerosol effects is at best questionable with even very large sample sizes. Thus, updraft speed proxies are typically used, most commonly consisting of cloud top height or temperature, radar reflectivity profile, or lightning flash rate. However, these proxies do not necessarily require a change in updraft speed to change. This is particularly true for lightning and reflectivity that correspond directly to microphysical changes typically associated with riming. Thus, maximum cloud top height, minimum cloud top temperature, or radar echo top as estimated via satellite or radar observations are more commonly used with the assumption that stronger updrafts will reach greater depths, at least partly because they may be warmer with a higher level of neutral buoyancy (LNB; also known as the equilibrium level), which is the height or temperature at which the convective updraft switches from positive to negative buoyancy. A further difficulty is that convective cloud system macrophysical properties vary in space and time due to growth, decay, and aggregation, often inclusive of an ensemble of updrafts within the single cloud system. For a given event, a spectrum of updraft speeds is expected due to variable updraft widths, cloud base thermodynamic conditions, near cloud thermodynamic and wind conditions, and entrainment-driven dilution created by convective and mesoscale variability. An example of this variability is shown in Figure 2 for most unstable convective available potential energy (CAPE; assuming pseudo-adiabatic parcel ascent) and several different updraft strengths. The many shortcomings of observational proxies and their representativeness limit the robustness of aerosol-convection correlations.



Incorrect interpretation of such correlations is often caused by insufficient control for meteorological covariability with aerosols. Of critical importance is controlling for the factors most likely to modulate convective cloud top height and updraft speed, including those shown in Figure 3. For studies using cloud top proxies for updraft speed, it is critical to constrain LNB using a lifted parcel in an environmental thermodynamic profile, e.g., via pseudo-adiabatic or moist adiabatic ascent, but this is rarely done in observational studies of aerosol invigoration of convection. Although some recent studies consider CAPE, which provides an estimate of potential updraft strength that typically assumes pseudo-adiabatic ascent absent buoyancy dilution, pressure perturbation, and condensate loading effects, it was neglected along with LNB in foundational studies including Koren et al. (2005, 2010) and Li et al. (2011). Indeed, Varble (2018) showed that LNB and CAPE correlations with CN concentration caused the correlation of CN with convective cloud top height in the widely cited Li et al. (2011) study, which erroneously concluded that the correlation was due to aerosol cold-phase invigoration of deep convection. Similarly, the tropical eastern Atlantic region chosen in Koren et al. (2010) sits on a sharp climatological gradient in AOD, which increases from south to north along with deep convection and rainfall moving from a suppressed shallow trade cloud regime into the Intertropical Convergence Zone with greatly differing meteorological conditions, such as CAPE and LNB. In the case of Fan et al. (2018) exploring warm-phase invigoration, CAPE was concluded to be similar across all aerosol conditions, but Öktem et al. (2023) showed that the CAPE computations were corrupted by bad surface data in soundings, sampled at different times during the day over land where CAPE has a strong diurnal cycle, and tended to be lower in the low-aerosol concentration conditions. Thus, in addition to including key meteorological variables in analyses, studies need to ensure that they are accurately measured and representative.

Once meteorological variables are chosen for analysis, a statistical approach is required to control for their correlations with aerosols. Many studies have attempted to do this by separating data into atmospheric state bins, but this has been shown via modeling to poorly control for such effects (e.g., Boucher and Quaas 2013). The failure of this approach results from mixing cloud and meteorological regimes together that do not retain the most important factors for a specific type of cloud. It also results from small changes in some factors such as water vapor mixing ratio having large impacts on cloud properties relative to those potentially caused by large changes in aerosol concentrations (Varble 2018). Such binning approaches also ignore the simultaneous impacts of many factors on convective updraft strength that require control via multiple linear regression, random forest, or other techniques that use all predictors as input together. In addition to CAPE and LNB, other meteorological factors likely to impact updraft strength are boundary layer depth, lifted condensation level, mid-level RH, and vertical wind shear, all of which can correlate with aerosol concentration. There are also many processes that modulate convective updraft strength including entrainment and condensate fallout (Figure 3). As Figure 3 shows, some of these processes are likely impacted by aerosol loading. Many more processes that impact updraft speed have unknown relationships with aerosol loading, particularly as convective cloud complexity increases with inclusion of ice processes and mesoscale organization. There are many complex process pathways extending beyond warm- and cold-phase invigoration for aerosols to correlate with and/or affect updraft strength positively or negatively, most of which have been neglected in past observational studies.





Sampling of representative meteorological conditions is similarly difficult to sampling CCN concentration and updraft speed, where near-cloud conditions including the inflow to the updraft are often not sampled. Thus, conditions from 390  discrete and often distant radiosonde measurements or reanalyses are commonly used instead. This introduces uncertainty because meteorological conditions often have substantial mesoscale variability along with aerosol conditions, as shown in Figure 2. Such variability has been observed by dense radiosonde networks where the low-level water vapor mixing ratio has been shown to vary by several g/kg on scales of 1 hour and 30 km in deep convective conditions (Nelson et al. 2021). This means that representativeness errors can be substantial, requiring large sample 395  sizes to overcome. Figure 2 highlights another complicating factor, which is precipitation scavenging of aerosols and stabilizing of the atmosphere in cold pools that form beneath and extend laterally outward from deep convection. Sampling of cold pool contaminated air will lead to aerosol-meteorology-cloud correlations that can be misinterpreted as aerosol effects on clouds. The effects of precipitation-reduced aerosols and stabilized air can persist for many hours and depend on the timescales for aerosol and instability recovery. Varble (2018) showed that a positive correlation 400  between aerosol concentration, LNB, and CAPE was related to the amount of precipitation that occurred earlier, setting up a situation in which cloud effects on aerosols can be incorrectly interpreted as aerosol effects on clouds.

Controlling for cloud conditions is often as important as controlling for meteorology. Koren et al. (2005, 2010) and many subsequent studies have combined purely liquid and mixed-phase clouds, while attributing aerosol-cloud top correlations to cold-phase invigoration. Others such as Li et al. (2011) have assumed that cloud tops colder than a 405  threshold such as -4°C contain ice. However, Varble (2018) showed that cloud tops < -4°C considered in Li et al. (2011) were bimodal with a congestus mode below -10°C that was likely purely liquid and another at < -55°C that represented the deep convection mode. Removing the congestus mode in that case removed any correlation between CN and cloud top height. Thus, some of the foundational observational studies supporting cold-phase invigoration in fact were showing correlations likely dominated by liquid clouds. Beyond separating warm and cold clouds, how 410  clouds are sampled can bias results. For example, excluding areas with 100% cloud fraction points in satellite analyses as done for 1° x 1° regions in Koren et al. (2005, 2010) removes large portions of MCSs, a bias that increases as MCS size increases. Indeed, cloud fraction is frequently positively correlated with cloud depth and attributed to aerosol effects, as in Koren et al. (2005, 2010), but mixing of different cloud types and meteorological conditions with analyses that do not consider entire cloud systems as individual entities can cause such correlations. Multiple sampling of 415  individual convective systems can also result in dependent sampling that erroneously increases statistical significance and biases samples to relatively large and long-lived systems. Sampling from a limited field-of-view instrument, such as a vertically profiling radar, can create similar sampling biases and additionally include highly unrepresentative samples. Indeed, Öktem et al. (2023) showed this was the case in Fan et al. (2018) by comparing to more representative sampling from a scanning precipitation radar. Thus, cloud sampling choices and impacts need to be carefully 420  considered when interpreting analyses.

Inappropriate statistical analyses are another common failure point. Several studies (Bell et al. 2008, 2009; Rosenfeld and Bell 2011) concluded that increasing aerosol loading enhanced convective depth, precipitation, hail, and tornadoes in portions of the south-central and/or southeastern U.S. based on a weekly cycle in these parameters and particulate



matter with a peak during the week and a lull on weekends. However, Kim et al. (2010) showed that robust regional
weekly cycles could emerge in the same region from natural meteorological variability, even when using 60 years of
data. Daniel et al. (2012) further showed how spatial and temporal autocorrelation coupled with an inappropriate usage
of the Student t-test produced spurious significance in weekly cycles. They also pointed out problematic post hoc
selection of analysis regions and time periods based on the presence of weekly cycles or not, something done in the
aforementioned studies that ignored regions where weekly cycles were absent. Other problems included not
accounting for correlations between atmospheric parameters and accepting a post hoc causal mechanism that could be
adjusted to be consistent with a range of results that can just as easily emerge from random variability or confounding
factors. Yuter et al. (2013) additionally showed how selective sampling in space and time that fit a particular
hypothesis while ignoring other possible explanations could lead to non-robust results and/or weakly supported
interpretations of causal mechanisms.

Many studies of the last decade claiming aerosol invigoration of deep convection as the source for correlations between
aerosol and deep convective cloud properties heavily rely on the validity of conclusions acquired in the above
discussed foundational studies. These studies used questionable and sometimes faulty methods to support warm-
and/or cold-phase invigoration hypotheses without sufficient consideration of alternative explanations. When
combined with weaknesses in theoretical and modeling foundations, a conclusion of net invigoration of deep
convective updrafts via warm- or cold-phase pathways is highly questionable.

**5 A Path Forward**

Theoretical, modeling, and observational studies that serve as foundations for aerosol invigoration of deep convection
are often cited an order of magnitude more than follow-up studies showing critical flaws in their approaches and/or
interpretations. This has led to numerous later studies applying warm- or cold-phase invigoration pathways as
explanations for correlations in observational and modeling datasets without process-level evidence or consideration
of alternative explanations, frequently with methodological flaws that follow unsound approaches of earlier studies.
Many of these studies, often in "high impact" journals, fail to adequately describe uncertainties, provide caveats, and
supply enough information to be fully reproducible. With clear deficiencies in the seminal studies that underpin
arguments of aerosol invigoration of deep convection, what is the path forward for science on this and related topics?

A critical first step is clarifying what is meant by aerosol invigoration of deep convection. The warm- and cold-phase
invigoration pathways contend that invigoration means an increase in updraft speed. However, many studies conflate
this definition with changes to microphysical properties that affect other aspects of deep convective clouds, such as
reflectivity, precipitation, and lightning. Such properties can change with aerosol concentration *without* a necessary
change in updraft strength. A critical second step is to estimate the expected magnitudes of such effects across different
atmospheric and cloud conditions so that proper observational and modeling approaches can be designed to isolate an
effect of that magnitude. For example, if a net 1% change in the convective updraft strength is expected for a doubling
of the CCN concentration in a particular meteorological setting, what accuracy is needed in the observational
estimation of meteorological, aerosol, and cloud properties to robustly isolate that effect, and how many independent




samples are needed? Some studies, such as Grabowski (2018) and Lebo (2018), have made first attempts to answer
this question, and more studies are needed to build on their findings. Such information underpins the statistical
methods required to achieve robust results, methods that have often fallen short in many past studies due to selective
sampling, a lack of proper control for confounding factors including meteorology and cloud effects on aerosols, and
little consideration for alternative explanations. To counter methodological flaws and avoid questionable conclusions
of past observational studies, we recommend the following approaches:

1. Continue improving CCN, convective updraft, and atmospheric state retrievals, and consider the impacts
from deficiencies of CCN and convective updraft strength proxies used in analyses.

2. Isolate single convective cloud types (e.g., purely liquid vs. mixed phase) and assess the representativeness
of aerosol, cloud, and meteorological sampling times and locations.

3. Avoid post-hoc or subjective selections of sampling times and regions that fit a preconceived narrative.

4. Control for atmospheric state parameters known to modulate the convective strength proxy (e.g., LNB for
cloud top height) by performing multivariate analyses that account for covariabilities between *all* predictor
variables.

5. Apply appropriate significance testing accounting for dependent sampling and non-parametric distributions.

6. Avoid adopting explanations for aerosol-cloud relationships from previous studies without evidence that such
explanations are more likely than possible alternatives.

Modeling-based conclusions related to deep convection invigoration by aerosols have also often been questionable.
To improve confidence in model-derived sensitivities of deep convective clouds to aerosols, we recommend the
following:

1. Continue improving the representation of updraft dynamics and microphysics in numerical models.

2. Expand the usage of LES to limit biases associated with under-resolved deep convective updrafts.

3. Assess the robustness of results with initial/boundary condition ensembles, simulations across different
convective regimes, and model intercomparisons.

4. Consider the limitations of chosen boundary conditions, time integration, domain size, and physics
parameterizations in application of findings to the real world.

5. Use objective and representative sampling methods to evaluate model output.

6. Provide observational context to assess confidence in model-derived sensitivities.

The community also needs to wrestle with prioritization of efforts and where investments will be potentially most
impactful given the many shortcomings of current observations and modeling to address. Supersaturation is of critical
importance to warm-phase invigoration, but values of supersaturation in updrafts remain uncertain. Thus, expanded
quasi-equilibrium supersaturation retrievals and evaluation of their validity across a range of updraft strengths, cloud
life cycle stages, and ambient environments are one area to focus efforts. Cold-phase invigoration depends on
condensate loading and freezing depths, two factors that are highly variable and could be better quantified with
targeted measurements and modeling as a function of updraft and cloud properties. Further, because measurements
within deep convective updrafts will always be limited, efforts could target creative ways to infer updraft properties
from remote sensing observations using, for example, observational simulators applied to LES. Such observations will



be critical for model evaluation and promoting continued model improvement that is required for accurate quantification of aerosol-deep convection interactions. Lastly, within the realm of aerosol-deep convection interactions, there is a case to be made that aerosol dynamical invigoration of convection has received an outsized research focus over potentially larger magnitude and more impactful direct aerosol effects on microphysical properties
that modulate convective precipitation and cloud radiative effects in ways that are not well understood. Whatever subsequent research into aerosol effects on deep convection is performed, it behooves the community to be mindful of methodological limitations and alternative explanations for findings while avoiding non-evidence-based conclusions that depend solely on previous studies.

**Author Contribution**

AV, AI, HM, WG, ZL: conceptualization. AV, HM, AI, WG: writing – original draft preparation. AV, HM, ZL, AI, WG: writing – review and editing. AV, AI: visualization. AV: supervision.

**Competing Interests**

The authors declare that they have no conflict of interest.

**Acknowledgements**

This work was funded by the U.S. Department of Energy's (DOE) Office of Science Biological and Environmental Research as part of the Atmospheric System Research (ASR) program. Additional funding support was provided by DOE ASR grant DE-SC0022149 (ALI), DOE ASR grants DE-SC008648, DE-SC0016476, and DE-SC0020118 (HM and WWG), and NSF grant 2326943 (ZJL). Additional support for HM was provided by DOE ASR DE-SC0020104. Pacific Northwest National Laboratory is operated by Battelle for the DOE under Contract DE-AC05-76RLO1830.
Thank you to Zhixiao Zhang for performing the simulation used for Figure 2 with support from NSF grant 1661662, the NCAR Computational and Information Systems Laboratory, and the University of Utah Center for High Performance Computing. Computing support was also provided by the National Energy Research Scientific Computing Center, a DOE Office of Science User Facility supported by the Office of Science of the U.S. Department of Energy under Contract DE-AC02-05CH11231. Thank you also to James Marquis and Jerome Fast for feedback as
well as Cortland Johnson and Nathan Johnson for PNNL Creative Services support in generating Figure 3. Lastly, a special thanks to many colleagues with whom valuable discussions helped inform our opinions expressed in this paper.



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

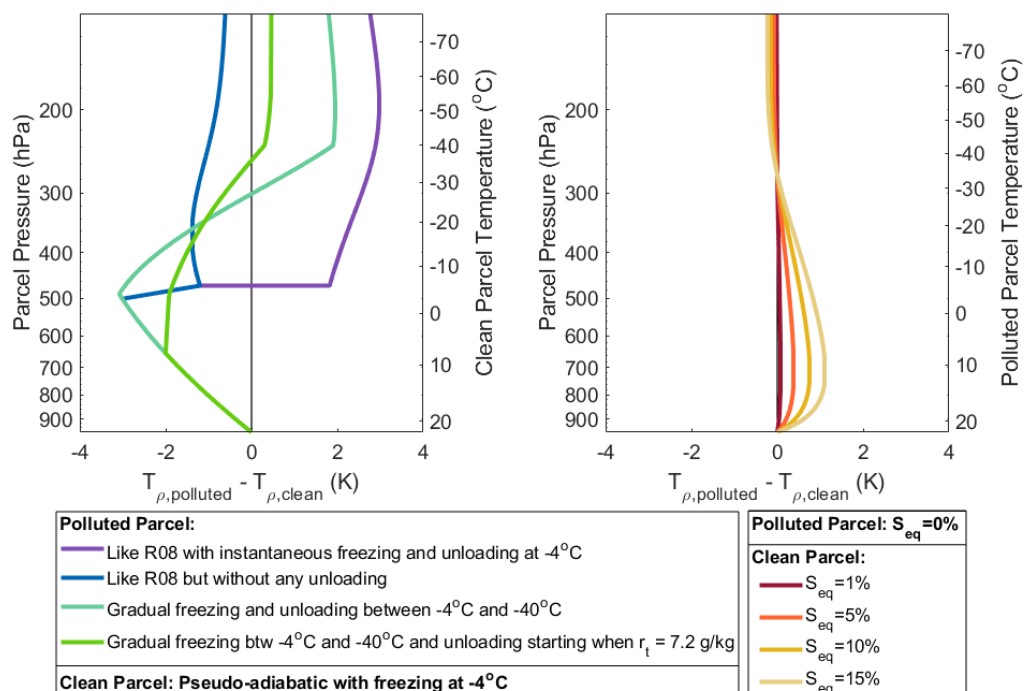


**Figure 1: Convective parcel calculations following Igel and van den Heever (2021) showing vertical profiles of density temperature for a polluted parcel relative to a clean parcel. (left) Following R08, the clean parcel is assumed to rise pseudo-adiabatically and carry no condensate while polluted parcels are shown for 4 different assumptions that become less idealized moving from purple to green. (right) The polluted parcel is assumed to maintain a supersaturation of 0% relative to liquid with 4 different equilibrium supersaturation values for the clean parcels shown.**


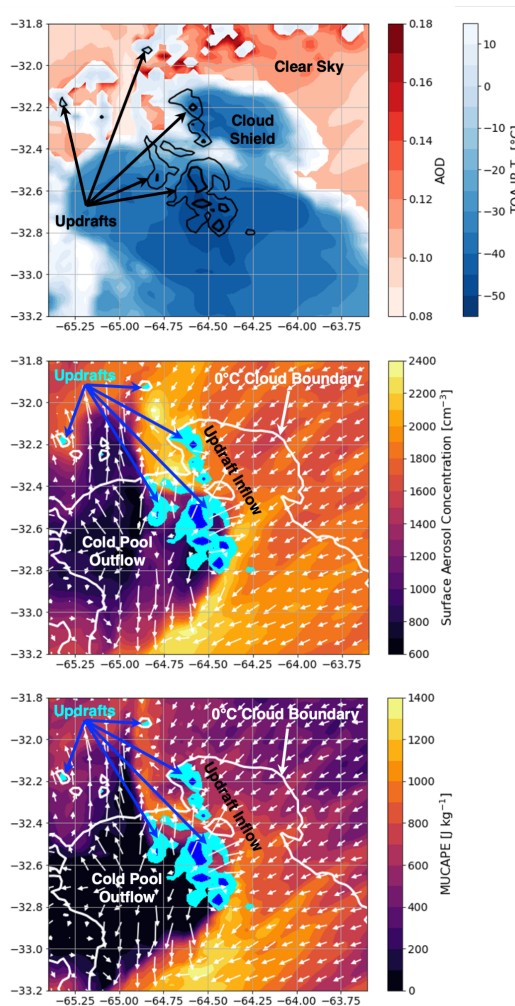

**Figure 2: An example 1.8° x 1.4° region in a 3-km WRF simulation (simulation details in Zhang et al. 2021) of deep convection highlighting complications with choosing a discrete location to observationally sample key atmospheric conditions that influence aerosol-updraft relationships. Examples include: (top) aerosol optical depth, top-of-atmosphere infrared brightness temperature (TOA IR $T_b$), and black contours of column-maximum vertical wind speed exceeding 3 and 9 m s$^{-1}$, (middle) surface aerosol concentration, and (bottom) most unstable CAPE (MUCAPE). Surface wind vectors are shown in (middle-bottom) with the 0°C top-of-atmosphere (TOA) infrared (IR) $T_b$ contour (white) and column maximum vertical wind speed exceeding 3 (cyan) and 9 (blue) m s$^{-1}$.**



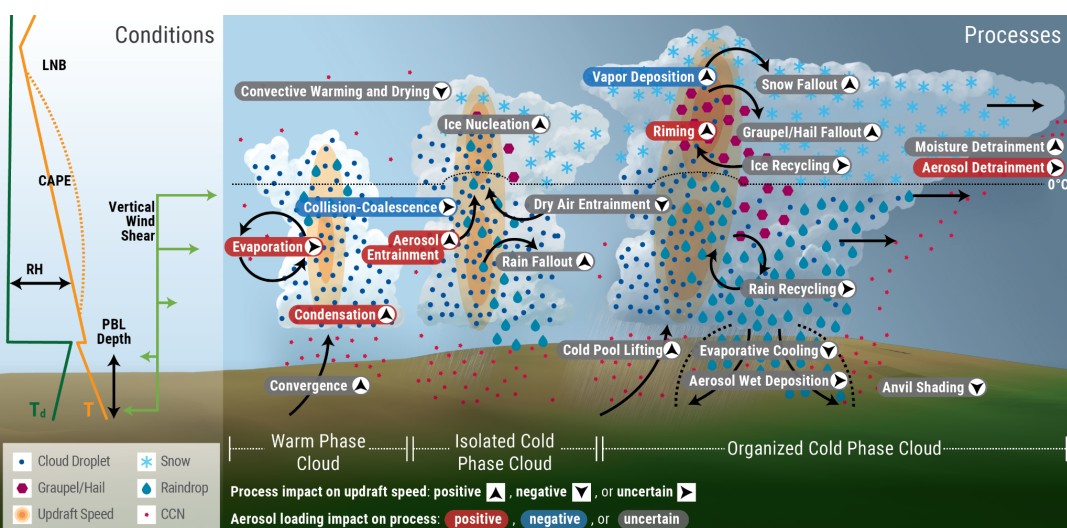

**Figure 3: Key atmospheric conditions and processes that modulate convective cloud updraft speed and depth in warm-phase, isolated cold-phase, and organized cold-phase convective clouds. Text coloring indicates the net impact of aerosol loading on a process and the arrow color indicates the net impact of a process on updraft speed based on the best judgments of the authors and studies to date with an acknowledgement that the sign of impacts can be variable. Gray colors indicate uncertain net impacts. Although processes are shown for specific cloud types, liquid and out-of-cloud processes apply across all cloud types, while ice processes apply for both isolated and organized cold clouds, though with greatly varying levels of importance. Note that uncertain impacts increase from left to right as cloud complexity increases, which highlights the difficulty in assessing overall aerosol effects.**