# Peer review of "Opinion: A Critical Evaluation of the Evidence for Aerosol Invigoration of Deep Convection"

_EGUsphere, 2023_

## Referee Comment (RC1)

Reviewer of:

**Opinion: A Critical Evaluation of the Evidence for Aerosol Invigoration of Deep Convection**

This paper evaluates the theoretical, modeling, and observational evidence for warm- and cold-phase invigoration pathways in deep convective clouds. It does a commendable job of reviewing previous work and highlighting that previous papers presenting evidence/explanations for convective invigoration are, at least partially, based on inaccurate assumptions, data sampling strategies, and statistical methods. Consequently, it claims that foundational observational studies supporting convective invigoration are highly questionable and provides supporting arguments. The authors also offer suggestions for a way forward.

I believe this opinion paper will make a significant contribution. Science should be driven by questioning previous results and raising doubts whenever alternative explanations for trends in observations or modeling data are possible. Moreover, I think this opinion paper is timely, considering the substantial body of literature published in recent years (mostly by the authors) that has questioned previous results. Additionally, this paper is well-written. While I think that the paper is in a publishable form as is, below I list a few minor suggestions that authors might want to consider.

**Comments:**

Line 54: I suggest listing the major relevant papers here instead of referencing Igel and van den Heever, 2021 to make it easier for the reader.

Line 55: I agree that a clear definition of "invigoration" is required, and focusing on vertical velocity makes the most sense. However, there are cases where an aerosol perturbation leads to larger mass flux to the upper troposphere, even without a change in vertical velocity, which subsequently affects cloud macro-physical and radiative properties (Dagan et al., 2020). It might be worth mentioning this in relation to the climate-relevant part in line 60.

Line 100: I suggest adding the paper by Marinescu et al., 2021, which also demonstrates diverging trends in multi-model comparison.

Line 121: This statement strongly depends on how one defines "notable." Wouldn't a 2-3% super-saturation under clean conditions be enough to make an impact? Romps et al., 2023 puts the limit at >1%. Additionally, while I appreciate the intention to present a similar x-axis range in Fig.1 left and right panels, I find the range presented in the right panel somewhat misleading. A 1K difference is already quite large, and the current presentation of this figure gives the impression that it is very small.

Line 134: I agree that warm-phase *invigoration* cannot be negative, but the aerosol effect on warm-phase convection can be negative (Jiang et al., 2006; Small et al., 2009; Dagan et al., 2015).

Line 412: The sentence starting with "Indeed" is a bit long and complicated to follow. I suggest splitting it into two parts.

Around line 480: I would add another point here: "try to avoid making strong conclusions based on a single model simulation" or "focus more on model intercomparisons." These MIPs have proven to be very informative in different sub-disciplines (e.g., RCEMIP for convective self-aggregation and cloud feedback), and I think they are not utilized enough in the ACI community.

**References:**
Dagan, G., Koren, I., and Altaratz, O.: Competition between core and periphery-based processes in warm convective clouds – from invigoration to suppression, Atmospheric Chemistry and Physics, 15, 2749-2760, 2015.
Dagan, G., Stier, P., Christensen, M., Cioni, G., Klocke, D., and Seifert, A.: Atmospheric energy budget response to idealized aerosol perturbation in tropical cloud systems, Atmospheric Chemistry and Physics, 20, 4523-4544, 2020.
Jiang, H., Xue, H., Teller, A., Feingold, G., and Levin, Z.: Aerosol effects on the lifetime of shallow cumulus, Geophysical Research Letters, 33, 10.1029/2006gl026024, 2006.
Marinescu, P. J., Van Den Heever, S. C., Heikenfeld, M., Barrett, A. I., Barthlott, C., Hoose, C., Fan, J., Fridlind, A. M., Matsui, T., and Miltenberger, A. K.: Impacts of varying concentrations of cloud condensation nuclei on deep convective cloud

updrafts – a multimodel assessment, Journal of the Atmospheric Sciences, 78, 1147-1172, 2021.

Romps, D. M., Latimer, K., Zhu, Q., Jurkat-Witschas, T., Mahnke, C., Prabhakaran, T., Weigel, R., and Wendisch, M.: Air pollution unable to intensify storms via warm-phase invigoration, Geophysical Research Letters, e2022GL100409, 2023.

Small, J. D., Chuang, P. Y., Feingold, G., and Jiang, H.: Can aerosol decrease cloud lifetime?, Geophysical Research Letters, 36, 2009.

---

## Author Comment (AC1)

We thank reviewers for their thorough review of our manuscript and believe comments have led to revisions that have improved its quality. Please find our responses and revisions based on them below in blue. All line numbers in red refer to changes made in the revised manuscript that includes tracked changes.

A few minor edits not based on the reviewer feedback below were made based on email feedback from a couple members of the research community on lines 110, 124, 137-140, 206, 361-362, 381-382, 486-488, and 544-546.

Reviewer 1

This paper evaluates the theoretical, modeling, and observational evidence for warm- and cold-phase invigoration pathways in deep convective clouds. It does a commendable job of reviewing previous work and highlighting that previous papers presenting evidence/explanations for convective invigoration are, at least partially, based on inaccurate assumptions, data sampling strategies, and statistical methods. Consequently, it claims that foundational observational studies supporting convective invigoration are highly questionable and provides supporting arguments. The authors also offer suggestions for a way forward.

I believe this opinion paper will make a significant contribution. Science should be driven by questioning previous results and raising doubts whenever alternative explanations for trends in observations or modeling data are possible. Moreover, I think this opinion paper is timely, considering the substantial body of literature published in recent years (mostly by the authors) that has questioned previous results. Additionally, this paper is well-written. While I think that the paper is in a publishable form as is, below I list a few minor suggestions that authors might want to consider.

Thank you for the encouraging comments.

**Comments:**

Line 54: I suggest listing the major relevant papers here instead of referencing Igel and van den Heever, 2021 to make it easier for the reader.

We have added major relevant papers to the revised paper on lines 56-57.

Line 55: I agree that a clear definition of "invigoration" is required, and focusing on vertical velocity makes the most sense. However, there are cases where an aerosol perturbation leads to larger mass flux to the upper troposphere, even without a change in vertical velocity, which subsequently affects cloud macro-physical and radiative properties (Dagan et al., 2020). It might be worth mentioning this in relation to the climate-relevant part in line 60.

This is a good point. We have clarified on lines 66-68 that the convective mass flux is also climatically important and not solely controlled by the vertical velocity; thus, aerosol effects on other aspects of convection such as its areal coverage are also of interest.

Line 100: I suggest adding the paper by Marinescu et al., 2021, which also demonstrates diverging trends in multi-model comparison.

We have added this citation on line 121.

Line 121: This statement strongly depends on how one defines "notable." Wouldn't a 2-3% super-saturation under clean conditions be enough to make an impact? Romps et al., 2023 puts the limit at >1%. Additionally, while I appreciate the intention to present a similar x-axis range in Fig.1 left and right panels, I find the range presented in the right panel somewhat misleading. A 1K difference is already quite large, and the current presentation of this figure gives the impression that it is very small.

At which point a supersaturation change is sufficiently significant is indeed arbitrary, and this point deserved further discussion. We have added some text on lines 146-150 discussing how the percentage that matters depends on the updraft being considered in terms of the impact of an absolute change in buoyancy to updraft speed. We have also reduced the x-axis range in the right panel of Figure 1.

Line 134: I agree that warm-phase *invigoration* cannot be negative, but the aerosol effect on warm-phase convection can be negative (Jiang et al., 2006; Small et al., 2009; Dagan et al., 2015).

This is a great point. We had originally left this out because studies primarily focus on shallow convection, and it isn't clear how relevant such effects are for deep convection. Nonetheless, it is possible that such entrainment-driven effects are relevant, and thus, we have added text discussing this on lines 167-172 and altered the text on line 166.

Line 412: The sentence starting with "Indeed" is a bit long and complicated to follow. I suggest splitting it into two parts.

Thanks for pointing this out. We have split this sentence in two on line 498.

Around line 480: I would add another point here: "try to avoid making strong conclusions based on a single model simulation" or "focus more on model intercomparisons." These MIPs have proven to be very informative in different sub- disciplines (e.g., RCEMIP for convective self-aggregation and cloud feedback), and I think they are not utilized enough in the ACI community.

Great point. We have altered the third point where we mention the robustness of results and model intercomparison to state to avoid strong conclusions based on a single simulation. The revised text is on lines 572-573.

**References:**

Dagan, G., Koren, I., and Altaratz, O.: Competition between core and periphery-based processes in warm convective clouds – from invigoration to suppression, Atmospheric Chemistry and Physics, 15, 2749-2760, 2015.

Dagan, G., Stier, P., Christensen, M., Cioni, G., Klocke, D., and Seifert, A.: Atmospheric energy budget response to idealized aerosol perturbation in tropical cloud systems, Atmospheric Chemistry and Physics, 20, 4523-4544, 2020.

Jiang, H., Xue, H., Teller, A., Feingold, G., and Levin, Z.: Aerosol effects on the lifetime of shallow cumulus, Geophysical Research Letters, 33, 10.1029/2006gl026024, 2006.

Marinescu, P. J., Van Den Heever, S. C., Heikenfeld, M., Barrett, A. I., Barthlott, C., Hoose, C., Fan, J., Fridlind, A. M., Matsui, T., and Miltenberger, A. K.: Impacts of varying concentrations of cloud condensation nuclei on deep convective cloud updrafts – a multimodel assessment, Journal of the Atmospheric Sciences, 78, 1147- 1172, 2021.

Romps, D. M., Latimer, K., Zhu, Q., Jurkat-Witschas, T., Mahnke, C., Prabhakaran, T., Weigel, R., and Wendisch, M.: Air pollution unable to intensify storms via warm- phase invigoration, Geophysical Research Letters, e2022GL100409, 2023.

Small, J. D., Chuang, P. Y., Feingold, G., and Jiang, H.: Can aerosol decrease cloud lifetime?, Geophysical Research Letters, 36, 2009.

**Reviewer 2**

**General Comment:** This opinion article summarizes the complexities of assessing the microphysical impacts of aerosol particles on clouds, both from modeling and observational perspectives. Overall, the article was well-written and nicely demonstrates many of the uncertainties associated with answering the question on whether cloud condensation nuclei concentrations strengthen convective cloud vertical motions. I provided some comments below that I think would make the manuscript clearer, as well as a few important points and citations that I think the authors have missed.

Thank you for your very helpful perspectives.

**Specific Comments:**
Title: How do the authors define deep convection? Since one of the two mechanisms that the authors are focusing on is the warm-phase mechanism, perhaps the title should be invigoration of "Convective Clouds" instead of "Deep Convection?"

We feel that generalizing to convective clouds including shallow convection would dilute the focus of the paper and that we would need to add more discussion on processes that are particularly relevant to shallow convection that have not been shown to necessarily be important for deep convection. Thus, we have chosen to stick with the focus on deep convection. We have added a definition for deep convection on lines 34-36 to clarify that buoyancy-driven clouds with updrafts extending from the lower to upper troposphere, often containing both liquid and ice phase regions, are the primary focus of the paper.

L34: The authors initially state that "There are many proposed effects of aerosol on deep convection," and then subsequently focus the remainder of the manuscript on how cloud condensation nuclei may impact hydrometeor production and related microphysical impacts. While the authors do state on L63 that this paper will focus how aerosol will impact latent heating, it would perhaps be useful to make very clear somewhere in these initial stages of the manuscript that the following topics are not discussed, even though they could also impact this invigoration question: 1) aerosol radiative effects and 2) that aerosol can act as ice nucleating particles, which can also impact the microphysics within clouds. To be clear, I do not think the authors should go into much detail on these other topics as they themselves are complex, but rather mention them as additional factors to this problem.

We agree this makes the manuscript objectives clearer. The first paragraph of the paper focuses on only microphysical modification, and thus, it seems out of place to mention dynamical effects there. However, the second paragraph transitions to dynamical effects and we have moved/added text to lines 52-58 to state the focus of the paper earlier than the third paragraph.

Ice nucleating particle (INP) effects on microphysics are mentioned in the first paragraph but not mentioned with respect to effects on convective dynamics. While INP effects on dynamics seems possible, we are not aware of a study that clearly lays out such pathways, and thus, we have left this discussion out of the second paragraph.

L100: Marinescu et al., 2021 also shows many models with negative impacts on updrafts in the cold-phase regions (above 7km AGL) with the ensemble median having no response to aerosol in this region and should be included here as well.

Thanks. We have added this citation on line 121.

L170: One of the primary findings of Marinescu et al., 2021 is that the consistent warm phase invigoration in 7 different models is likely attributed to increased environmental instability due to aerosol-induced boundary layer changes and can be included here as another example.

We added this example on lines 213-215.

L230-232: Many additional studies have shown an assessment of updraft changes under different environmental conditions, so I think the citation list on L232 should have an e.g. in front of it. Some additional examples of the early research on this, as well as some more recent work are Fan et al., 2007; Lee et al., 2008; Storer et al., 2010; and Sokolowsky et al., 2022, but there are many others as well.

We agree. In many cases throughout the paper, there were too many relevant studies to list, so we were selective in highlighting specific examples. We reviewed the additional studies listed and agree that they fit as additional examples that complement the others already listed. We have added these on lines 281-283.

L234-238: I found this statement a little confusing, and perhaps the readership would benefit from a clearer explanation here. Specifically, how can an environment "adjust to aerosol-induced convective invigoration"?.

The simplest example is that convection acts to heat the environment (via latent heating) and dry the environment (via precipitation). If the heating and drying is enhanced via convective invigoration, then the environment will stabilize more due to that heating and drying, i.e., the convection is working to remove convective instability. This means that other convective clouds in the area can be suppressed due to enhanced stabilization. This is particularly true if the large-scale forcing is fixed because it dictates the large-scale vertical motion and precipitation such that invigorated convection consumes more of the available potential energy, leaving less for the other clouds. Even when the large-scale forcing is not fixed, it is possible that the effects of invigorated convection at one location and time will suppress nearby or later convective clouds through the same effects. We have added some clarification on lines 287-289 that this is resulting from enhanced stabilization via heating and potential low-level drying (which also depends on other factors like precipitation efficiency).

L265. There have been community-wide model intercomparison studies to assess aerosol effects on clouds, with several more currently on-going. These model intercomparison projects have even focused on topics such as CCN invigoration of updrafts (Marinescu et al., 2021). I think these community-wide efforts should be highlighted here, as they also represent an ensemble of results and a way to understand variability in modeling results.

We agree that model intercomparisons are an important and separate method to assess robustness as compared to meteorological ensembles for a single model or piggybacking. We have added text on lines 322-325 to reflect this and have also clarified modeling recommendation #3 in section 5 (lines 572-573).

L277: Were "substantial biases" in model simulations discussed? Perhaps consider removing "biases" here or mention what these biases are, with references?

Good catch. We did not discuss biases apart from uncertainties. We now added some on lines 259-262.

L374-376: "The failure of this approach results from mixing cloud and meteorological regimes together …" Are the authors stating that the issue is that regimes need to be further constrained by both cloud types AND meteorological regime? I found this statement a little confusing.

This was confusing. We have attempted to clarify the argument being made here on lines 440-449.

L915: Should arrow "color" be arrow "direction?"

Yes. Thank you for spotting this. The caption for Figure 3 has been modified accordingly.

**References:**

Fan, J., R. Zhang, G. Li, and W. K. Tao, 2007: Effects of aerosols and relative humidity on cumulus clouds. *J. Geophys. Res.*, **112**, D14204, https://doi.org/10.1029/2006JD008136.

Lee, S. S., L. J. Donner, V. T. J. Phillips, and Y. Ming, 2008: The dependence of aerosol effects on clouds and precipitation on cloud-system organization, shear and stability. *J. Geophys. Res.*, **113**, D16202, https://doi.org/10.1029/2007JD009224.

Marinescu, P. J., and Coauthors, 2021: Impacts of Varying Concentrations of Cloud Condensation Nuclei on Deep Convective Cloud Updrafts—A Multimodel Assessment. *J. Atmos. Sci.*, **78**, 1147–1172, https://doi.org/10.1175/JAS-D-20-0200.1.

Sokolowsky, G. A., S. W. Freeman, and S. C. van den Heever, 2022: Sensitivities of Maritime Tropical Trimodal Convection to Aerosols and Boundary Layer Static Stability. *J. Atmos. Sci.*, **79**, 2549–2570, https://doi.org/10.1175/JAS-D-21-0260.1.

Storer, R. L., S. C. van den Heever, and G. L. Stephens, 2010: Modeling aerosol impacts on convective storms in different environments. *J. Atmos. Sci.*, **67**, 3904–3915, https://doi.org/10.1175/2010JAS3363.1.